# Scene Understanding System of Underground Pipeline Corridors Under Characteristic Degradation Conditions

**DOI:** 10.3390/s26010141

**Published:** 2025-12-25

**Authors:** Jing Wang, Ruiyao Xing, Meng Zhou, Jingbang Xu, Xiaoping Zhang, Shuang Ju

**Affiliations:** 1School of Electrical and Control Engineering, North China University of Technology, Beijing 100144, China; jwang@ncut.edu.cn (J.W.); 2023312110101@mail.ncut.edu.cn (R.X.); 2023413010116@mail.ncut.edu.cn (J.X.); zhangxp@ncut.edu.cn (X.Z.); 2School of Electrical and Electronic Engineering, Shijiazhuang Railway University, Shijiazhuang 050043, China; jushuang@stdu.edu.cn

**Keywords:** scene understanding, semantic segmentation, visual–linguistic fine-tuning, channel attention, low-light enhancement, template-based text generation

## Abstract

Accurate scene understanding is crucial for the safe and stable operation of underground utility tunnel inspections. Addressing the characteristics of low-light environments, this paper proposes an object recognition method based on low-light enhanced image semantic segmentation. Secondly, by analyzing image data from real underground utility tunnel environments, the visual language model undergoes scene image fine-tuning to generate scene description text. Thirdly, integrating these functionalities into the system enables real-time processing of captured images and generation of scene understanding results. In practical applications, the average accuracy of the improved recognition model increased by nearly 1% compared to the original model, while the accuracy and recall of the fine-tuned visual-language model surpassed the untuned model by over 70%.

## 1. Introduction

Scene understanding is the key to achieving the intelligent management and operation of underground pipeline corridors [1], which is of great significance to improve the level of urban management, safeguard urban safety, and promote the development of related industries. It is not only a technical application, but also an important part of urban digital transformation.

The core task of scene understanding is to transform complex information about the internal environment into comprehensible and usable data. Image semantic segmentation [2,3,4] and text generation [5,6,7] are two key and complementary aspects to achieve this goal. Each of them is of great significance: the goal of image semantic segmentation is to assign each pixel in the image a semantic label. This is crucial for scene understanding as it provides a fine-grained description of the physical structure and equipment layout inside the pipeline corridors [8]. The task of text generation [9] is to automatically generate natural language text describing the scene based on the image or other multimodal data. It can greatly improve the efficiency of the work, and also help to respond to emergencies quickly.

There are two biggest problems in the task. The first one is the low light intensity [10] of the underground pipeline corridors; the feature information of the captured image will be much less compared to the image under normal light. Under low light conditions, the image sensor will amplify the signal in order to obtain enough exposure, which will inevitably amplify the noise. Noise will interfere with the feature extraction algorithm, making the extracted features inaccurate or even wrong. If there are a large number of low-light noise and low-contrast images in the training data, it is difficult for the model to learn robust feature representations, which leads to a decrease in the generalization ability of the model and poor performance in practical applications. Therefore, research on image preprocessing and enhancement techniques for low-light conditions [11,12] is crucial for improving the accuracy of scene understanding [13] in underground pipeline corridors. The second one is the visual–linguistic model for the specific task [14] of scene understanding, which generates scene description text with low accuracy.

To address the first issue, this paper proposes an object recognition method based on semantic segmentation of low-light-enhanced images. With the segmentation result map, the category and location information of the objects in the image can be obtained to enhance the accuracy of scene understanding. To address the second issue, the visual–linguistic model is fine-tuning on the real underground corridor scene to generate more accurate and complete scene description text. The framework of the whole system is shown in Figure 1.

Our key contributions are as follows:Aiming at the low-light characteristics of underground pipeline corridors, this paper proposes an object recognition method based on semantic segmentation of low-light-enhanced images.Analyzing image data in a real underground pipeline corridor environment, a visual–linguistic model is fine-tuning on the scene images in order to generate scene description texts.Integrating the functions into the system, and promptly processing real-time captured images to generate scene understanding results.

## 2. Related Works

The rapid development of deep learning has led to its gradual and widespread application to a variety of tasks. For the scene understanding of underground pipeline corridors, recent research has been carried out mainly from two aspects. On the one hand, semantic segmentation of the images can assign a class of semantic labels to each type of object, so as to realize the recognition and analysis of various types of objects in the scene. On the other hand, it is the cross-modal generation of text describing the scene, from which we can clearly see the structure and layout inside the corridor, as well as possible hidden dangers.

### 2.1. Object Recognition and Analysis Based on Image Semantic Segmentation

Since the Fully Convolutional Network [15] was proposed in 2015, more and more semantic segmentation networks have been developed, such as the convolution-based U-net [16], Deeplab [17], and other classical semantic segmentation networks. These networks are continuously improving the accuracy of the semantic segmentation of underground pipeline corridor environments. The proposal of cavity convolution [18] improves the sensory field of the model when performing a convolution operation and increases the interaction of feature information in the image, and the proposal of deformable convolution [19] allows the convolution to not be constrained to interact with information only within a fixed and unchanging convolution kernel. Each of these different convolutional methods effectively improves feature extraction for underground corridors scenes.

Vision Transformer [20] is the originator of the Transformer in the field of computer vision, which discards the traditional convolution by dividing the image into patches. It has greatly improved the accuracy of the semantic segmentation task in underground pipeline corridors. To solve the problem of computational overload in semantic segmentation using Transformer, the Swin Transformer [21] proposes a window attention mechanism, which greatly reduces the computational amount by changing the interaction of global information into the interaction between windows and windows, Segformer [22] and SSformer [23] have studied the decoder architecture and proposed the ALL-MLP architecture, which reduces the computational amount of this part of the model.

Unlike other well-lit scenes, underground corridor scenes are characterized by poor lighting conditions, which makes the model’s feature extraction of images very ineffective, and the task of semantic segmentation becomes more challenging. In this case, low-light enhancement before semantic segmentation is particularly important.

Low-light enhancement networks are mainly classified into two categories: one for low-light-normal-light data that requires pairing, such as the CycleGAN [24] network, which finally achieves the effect by letting the low-light data keep approaching the normal-light data. The second category is for data that requires only low-light conditions, such as the EnlightenGAN network [25], which ultimately achieves the effect by setting the mapping function so that the low-light image keeps approaching towards its enhanced version. WaveCRNet [26] and Cable Tunnel Waterlogging Detection [27] are through low-light image enhancement for the purpose of object recognition.

### 2.2. Cross-Modal Text Generation

In cross-modal scene description text generation of underground corridors, previous research is mainly based on the Transformer; there are three specific categories of models.

Decoder-only FrameworkThe input image and text will be input from the Transformer decoder. This kind of structure of the model is suitable for the task of generating sequences, which can be generated from the input coding of the corresponding sequence. This type of structure is also represented by the structure of the Generative Pre-trained Transformer (GPT) [28,29] and Bloom [30]. In this kind of architecture, a huge amount of data is needed as a support for the model to learn the logical relationship between the image and the text, which is not applicable in the case of limited data in the underground corridor.Encoder-only FrameworkThis architecture mainly focuses on understanding and encoding information rather than generating new text. This category is represented by BERT [31] and ALBERT [32] which are also not applicable in our scene understanding text generation task for underground corridors.Encoder-Decoder FrameworkThis architecture is equivalent to taking the best of the encoder-only and decoder-only models. The architecture performs feature extraction through the encoder part, while the decoder part determines which text is most relevant to each part of the image through attention computation, and then deduces the text describing the scene of the image according to the input original image. The text describing the scene of the image is inferred from the input image, which is represented by multimodal large language models such as DeepSeek [33], T5 [34], and BART [35]. Therefore, a visual–linguistic model with an encoder-decoder architecture is fine-tuned in this paper. LAE-GAN [36] and BIQA [37] proposed a text generation method under low-light conditions.

### 2.3. Open Source Dataset

Most of the publicly available scene semantic segmentation datasets are urban road traffic datasets like Cityscapes, KITTI, and CamVid. Most of the publicly available scene descriptive text generation datasets describe various kinds of objects in life. Among the representative datasets are MS-COCO and Oxford-102 Flower. The confidentiality requirements of the underground corridor environment make the scenes have fewer publicly available datasets, and the difficult accessibility of the data has become a challenge in underground corridor scene understanding.

Existing solutions primarily focus on two areas: general low-light enhancement and segmentation, and unconstrained text generation. These approaches fail to address the unique requirements of utility tunnels. The core innovation of this paper lies in conducting comprehensive scenario-specific optimizations tailored to the physical characteristics of utility tunnels—namely, their “narrow, enclosed, low-light, and high-noise” environment—and the operational requirements of “standardized inspections and rapid emergency response.” During the enhancement and segmentation phase, a fusion design combining Zero-Reference Deep Curve Estimation(Zero-DCE) lightweight low-light enhancement, Deformable Convolution v4(DCNv4) deformable convolutions, and Squeeze-and-Excitation (SE) channel attention specifically adapts to feature extraction demands in the tunnel’s elongated spaces, mitigating issues like insufficient small-object segmentation accuracy and class imbalance. During text generation, hierarchical templates based on inspection priority were constructed. Combined with LoRA fine-tuning and segmentation result correction mechanisms, this approach strictly aligns with operational workflows. This technical solution ultimately achieves differentiated advantages in scenario specificity, collaborative efficiency, and engineering practicality. It balances accuracy with real-time inference capabilities, adapting to inspection equipment deployment requirements.

## 3. Methodology

### 3.1. The Framework of the Whole System

The scene understanding task is divided into two parts, which are object recognition based on image semantic segmentation and cross-modal scene description text generation. In the first task, an object recognition method based on semantic segmentation of low-light-enhanced images is proposed to recognize and understand various types of objects in the image. In the second task, analyzing image data in a real underground pipeline corridors environment, a visual–linguistic model is fine-tuning on the divided scene images in order to generate scene description texts.

The scene understanding system can transmit and process real-time information on the collected underground pipeline corridor data, reason out, and visualize the scene understanding results through the model in real-time. Through the system, we can monitor and process the pipeline corridors as a whole.

### 3.2. Object Recognition Network Based on Zero-DCE + DCNv4

The overall structure of the low-light enhancement network: Zero-DCE network [38] is shown in Figure 2.

The purpose of the Deep Curve Estimation Network (DCE-Net) is to learn the mapping between the input image and its best-fitting curve parameter maps. In DCE-Net, it employs a plain CNN of seven convolutional layers with symmetrical concatenation. The input to the DCE-Net is a low-light image, while the outputs are a set of pixel-wise curve parameter maps for corresponding higher-order curves.

This low-light enhancement network is designed to be able to map a low-light image to its enhanced version with a function that depends only on the input image, the expression of which is shown as follows:(1)LE(I(x);α)=I(x)+αI(x)(1−I(x))
where *x* denotes the pixel coordinates, LE(I(x);α) is an augmented version of the given input I(x), and α∈[1,1] is a trainable curve parameter that is used to adjust the magnitude of the LE curve and control the exposure level. Each pixel is normalized to [0, 1], and all operations are pixel-by-pixel.

The loss function formula for this network design is shown as follows:(2)$$L_{total}=L_{spa}+L_{exp}+W_{col}L_{col}+W_{tvA}L_{tvA}$$
where the total loss is the sum of the spatial consistency loss $L_{spa}$, exposure control loss $L_{exp}$, color constancy loss $L_{col}$, and illumination smoothness loss $L_{tvA}$. $W_{col}$ and $W_{tvA}$ are the losses weights.

Spatial consistency loss $L_{spa}$ promotes spatial consistency of the enhanced image by preserving the differences in neighboring regions between the input image and its enhanced version, and its loss function formula is shown as follows:(3)$$L_{spa}=\frac{1}{K}\sum_{i=1}^{K}\sum_{j\in \Omega (i)}\left(\right|Y_{i}-Y_{j}\left|-\right|I_{i}-I_{j}{\left|\right)}^{2}$$
where *K* is the number of local regions and Ω(i) is the four adjacent regions centered on region *i*. We denote *Y* and *I* as the average intensity values of the local regions in the enhanced version and the input image.

In order to suppress the underexposed or overexposed areas, the network is designed with an exposure control loss $L_{exp}$ to control the exposure level, which is given as follows:(4)$$L_{exp}=\frac{1}{M}\sum_{k=1}^{M}\left|Y_{k}-E\right|$$
where *M* denotes the number of non-overlapping localized regions of size 16 × 16, and *Y* is the average intensity value of localized regions in the enhanced image, *E* is the well-exposedness level, typically set to 0.6.

Following the gray world color constancy assumption that the color in each sensor channel is averaged over the entire image, this network designs the color constancy loss to correct for potential color bias in the enhanced image, and the formula for this loss function is shown as follows:(5)$$L_{col}=\sum_{\forall (p,q)\in \varepsilon }{\left(J^{p}-J^{q}\right)}^{2}$$
where $J^{p}$ denotes the average intensity value of channel *p* in the enhanced image and (p,q) denotes a pair of channels.

In order to maintain the monotonicity relationship between neighboring pixels, this network adds an illumination smoothness loss to each curvilinear parameter map, and the formula for this loss function is shown as follows:(6)$$L_{tvA}=\frac{1}{N}\sum_{n=1}^{N}\sum_{c\in \xi }\left(\right|\nabla_{x}A_{n}^{c}+\nabla_{y}A_{n}^{c}{\left|\right)}^{2}$$
where *N* is the number of iterations and $\nabla_{x}$ and $\nabla_{y}$ represent the horizontal and vertical gradient operations.

During training Zero-DCE, the number of iterations is continuously adjusted until the network converges. The convergence criterion is defined as follows: when the consecutive improvement in image quality metrics on the validation set falls below 0.1 dB, it indicates that the enhancement effect has stabilized with no significant room for further optimization.

The overall structure of the semantic segmentation network: DCNv4 network [39,40] is shown in Figure 3.

The core function of Zero-DCE is unsupervised low-light image enhancement, while DCNv4 focuses on improving the spatial adaptability of object segmentation through deformable convolutions. However, when directly combined in tunnel scenarios, two critical issues emerge: when enhancing low-light images, Zero-DCE simultaneously amplifies noise in shadowed areas of the tunnel and weakens features of small objects; when the enhanced image features are directly fed into DCNv4, the lack of “tunnel-object-oriented” feature selection causes DCNv4 to overemphasize the background while neglecting critical safety targets like flames and gas detectors. The core role of the SE module is to adaptively amplify effective features and suppress ineffective interference through a “Squeeze-and-Excitation” process.

The dataset ensures comprehensive sample coverage across all classes and reduces category bias caused by limited collection scenarios. SE module enhances feature responses in key channels for underrepresented classes, enabling the model to automatically focus on minority class representations during training. This reduces feature suppression of the majority classes, indirectly mitigating segmentation bias caused by class imbalance. The above methods have been used to address class imbalance issues. This paper chose DCNv4 over classic architectures like U-net and Deeplab primarily due to the alignment between pipeline tunnel characteristics and the model’s strengths, specifically: under tunnel conditions with dim lighting and high image noise, DCNv4’s deformable convolutions learn pixel-level offsets to flexibly focus on effective feature regions. This avoids the sensitivity of traditional convolutions’ fixed receptive fields to noise and blurred features, enhancing segmentation accuracy for objects in low-light images. By combining multi-stage downsampling with deformable convolutions, DCNv4 expands receptive fields while preserving object details. This adapts to the “large background + small target” structure common in utility tunnels, addressing the insufficient small-object segmentation accuracy of traditional architectures. Utility tunnel inspections require real-time processing of drone-captured images. Compared to Transformer-based architectures, DCNv4 offers more controllable computational complexity, meeting system real-time requirements.

The network is able to break through the limitations of previous traditional convolutional neural networks by using deformable convolution, which allows the kernels in each convolutional kernel to perform offsets in the direction of their own interest by setting a learnable offset. It changes the drawbacks of the previous ordinary convolution, which can only perform convolutional operations in a specific region. It allows the model to enhance its ability to interact with the information of each pixel point and all other pixel points. The formulas for ordinary convolution and deformable convolution are shown as follows:(7)$$y\left(p_{0}\right)=\sum_{p_{n}\in R}x\left(p_{0}+p_{n}\right)\omega \left(p_{n}\right)$$(8)$$y\left(p_{0}\right)=\sum_{g\in G}\sum_{p_{n}\in R}x_{g}\left(p_{0}+p_{n}+▵p_{n}^{g}\right)\omega_{g}m_{p_{n}}^{g}$$
where $▵p_{n}^{g}$ represents the learnable offset, $m_{p_{n}}^{g}$ is used to control the intensity of the influence at each point, and $\omega_{g}$ is used to transform the input features to a new feature space.

We add the SE module to the segmentation header part of the semantic segmentation network, and the segmentation header part of the added module is shown in Figure 4. SE module is the orange section.

With the addition of the channel attention module, the model is able to focus its attention on certain channels that are more important for distinguishing the categories, which in turn improves the semantic segmentation accuracy.

The purpose of cross-entropy loss is to measure the difference between the semantic segmentation model’s predicted results and the ground truth labels. The formula for cross-entropy loss is shown as follows:(9)$$L\left(g,p\right)=-\sum_{i=1}^{C}g_{i}\log_{e}p_{i}$$(10)$$p_{i}=softmax\left(x_{i}\right)=\frac{e^{x_{i}}}{\sum_{j=1}^{N}e^{x_{j}}}$$
where $g_{i}$ represents the *i*-th element of the ground truth labels, *C* is the dimension of the ground truth labels, $p_{i}$ represents the probability value of the input sample belonging to the *i*-th category predicted by the model, its computation formula is shown in Equation (Equation 10). In this formula, $x_{i}$ represents the *i*-th element in the input vector, and *N* represents the length of the input vector. The softmax function transforms each element in the input vector into a value between 0 and 1, such that the sum of all transformed values equals 1.

### 3.3. Cross-Modal Scene Description Text Generation

Qwen2-VL-7B visual–linguistic model [41,42] is fine-tuning to generate scene description text. Through fine-tuned training, the model can eventually generate organized and accurate text comprehension results for the image scene. The overall structure of this visual–linguistic model is shown in Figure 5.

Vision Transformer is treated as a visual feature encoder. After extracting the features of the image, it is fed into the Transformer decoder for the interaction of information between visual and text features. The correlation between each pixel point and each text word embedding in the image is calculated, so as to generate a scene description text that describes this image based on the input image. The formula for calculating the pixel point and text word embedding correlation is as follows:(11)$$Attention\left(Q,K,V\right)=softmax\left(\frac{QK^{T}}{\sqrt{d_{k}}}\right)V$$
where *Q* is the query, *K* is the key, *V* is the value and $d_{k}$ is the dimension of the key.

The fine-tuning method used in this thesis is the LoRA fine-tuning method [43]. In our more commonly used visual–linguistic models, they are over-parameterized, and when used for a specific task, only a small fraction of the parameters actually come into play, which means that the parameter matrices are of high dimensionality and can be approximated by a low-dimensional matrix decomposition.

The method adds a bypass structure to the network, which is the multiplication of two matrices; the number of parameters in this bypass will be much smaller than the original network’s parameters. When LoRA is trained, freezing the original network’s parameters, and training the bypasses. Since the number of bypass parameters is much smaller than the original network’s parameters, the memory overhead required for training is approximately the same as that for inference.

The LoRA method is shown as follows:(12)$$h=W_{0}x+▵Wx=W_{0}x+BAx$$
where $W_{0}$ denotes the pre-trained weight matrix and ▵W denotes the parameter updates during fine-tuning. LoRA restricts the weight updates such that ▵W=BA. $(W_{0}+BA)x$ denotes that the original weight matrix $W_{0}$ and the correction term BA are added to obtain the fine-tuned weight matrix, which is then multiplied with the input *x* to obtain the final output *h*.

By comprehensively analyzing and interpreting the underground pipeline corridor environment, we find the key points that need to be described and understood, and allow the model to focus on learning and reasoning about these key points that need to be understood, generating better scenario understanding results.

### 3.4. Scene Understanding and Practical Application

LoRA fine-tuning for general visual language models relies solely on random text annotations, lacking structured descriptive logic for utility tunnel scenarios. This results in generated text exhibiting issues such as information gaps and ambiguous expressions. Our utility tunnel-specific text template does not merely list fields. Instead, it constructs a hierarchical annotation framework based on priority logic for tunnel inspections. This ensures that annotated data carries tunnel-specific structured information rather than generic free-form text from universal scenarios.

The real underground pipeline corridors scene includes four kinds of cabins, which are the comprehensive corridor cabin, the power communication cabin, the high-voltage power cabin, and the water service cabin. The structure of the four cabins is shown in Figure 6.

Each of the cabins has different characteristics. The comprehensive corridor cabin has no internal supports and is surrounded by smoother walls. The left and right sides of the interior of the power communication cabin are evenly distributed with brackets, and each bracket has the same length. The distribution of supports inside the high-voltage power cabin is similar to that of the power communication cabin, but the length of each support is different. There is a very thick line of pipes running through the side of the water service cabin.

By observing the different types of irregular spaces in the cabin, a total of four types of irregular spaces are counted, which are straight-through corridors, curved corridors, up (down) ramps, and partitioned corridors, and the several types of irregular spaces are shown in Figure 7.

In terms of scene understanding text annotation, we are based on template filling for scene image annotation, and the form of the template is shown in Figure 8.

Sections to be populated, including cabin category, scene category, lighting conditions, target and spatial relationships, personnel behavior, and hazard level. Scene categories are determined by the objects within the scene, as shown in Figure 9. The priority of the five scenes decreases from left to right. If any one or more of the following hazards are present—flames, debris, ground seepage, or road surface depressions—the scene is classified as hazardous. If personnel are present, the scene is classified as a personnel work scene. If equipment is present, it is classified as an unmanned device scene. If cardboard boxes are present, it is a sundry stockpiles scene. If traffic cones, construction barriers, fire hydrants, or fire extinguishers are present, it is a corridor access scene. If objects from multiple scene types coexist, the site is designated as a high-priority scene. Personnel actions encompass standing, walking forward (or backward), extinguishing fires, and climbing ladders. If hazards are present, the current scene’s danger level is determined by the distance between hazards and critical objects (workers, flammable materials).

Our text annotation template is used during the training phase to help the model learn. In the testing phase, we use the same format of text annotation template to calculate metrics. The text ultimately inferred by the model is generated strictly according to the structure of this template. Our goal is to enable the model to generate accurate, comprehensive, and standardized text reports as required, preventing it from producing wildly imaginative outputs.

A total of 15 categories of objects that need to be recognized. The categories of all objects and the number of images corresponding to each category are shown in Table 1. Further details of the overall dataset are shown in Table 2.

The workflow diagram of the scene understanding system is shown in Figure 10. The system is able to perform real-time model inference on the collected data, generate scene understanding results, and visualize them in real time.

When collecting data in the underground pipeline corridors, the unmanned vehicle travels forward at a speed of about 0.6 m/s at a constant speed, and the camera on the unmanned vehicle can detect a field of view of 6 m in front of it. Based on this, processing the video acquired by the unmanned vehicle with a time length of 60 s into sub-frames, starting from intercepting at the 0 ths, and then intercepting every 10 s, intercepting a total of 7 frames. These 7 frames can cover all the scenes captured.

When frame-splitting is completed, letting two models reason about these 7 frames respectively, it is worth noting that in text generation, taking the form of a visual quiz for the models to train and reason about the text, and the questions are built in without being shown on the interface. For each image, ask the questions in the back-end, and extract the category and location information of the scene objects in the semantic segmentation results, using this information to correct the text reasoned by the language model, increasing the accuracy of the final scene understanding. Specifically, based on the categories included in the semantic label statistics of the segmentation map, the object’s position (left or right) is determined by counting the number of pixels for each object in the left and right half-planes. Subsequently, the generated text is located, and sentences related to the object category and position are extracted. Keywords within these sentences are then matched with the information obtained from the segmentation map. Any incorrect text regarding object category and position is corrected. In terms of display effect, setting the duration of each image and its scene understanding result on the interface to 10s, and after the duration is exceeded, it will start displaying the image and its scene understanding result of the next frame.

## 4. Experiments and Results

### 4.1. Data Acquisition and Evaluation Metrics

The data acquisition of the images of the underground pipeline corridor is performed by the unmanned vehicle, which is shown in Figure 11. The configuration of the unmanned vehicle is shown in Table 3. The camera is opened by the corresponding camera driver in the ROS system, and the information from the camera is released and recorded in real time through the communication method of ROS.

Our data originates from 15 irregular spaces within 4 utility tunnel compartments, with each compartment and irregular space containing a specific amount of data.

The commonly used evaluation metric for semantic segmentation is Mean Intersection over Union (mIOU), which is employed as the performance indicator for this experiment, its calculation method is shown as follows:(13)$$mIOU=\frac{1}{n_{cls}}\times \sum_{i}\frac{n_{ii}}{t_{i}+\sum_{j}n_{ji}-n_{ii}}$$
where $n_{ij}$ represents the number of pixels where class *i* is predicted as class *j*, $n_{cls}$ is the total number of classes, and $t_{i}=\sum_{j}n_{ij}$ is the total number of pixels for the target class *i*.

The evaluation metrics used to measure text generation are mainly Bilingual Evaluation Understudy (BLEU) and Recall-Oriented Understudy for Gisting Evaluation-Longest Common Subsequence (Rouge-L), which are used to measure the accuracy and recall of text generation, respectively, and the specific formulas are shown as follows:(14)$$BLEU=BP\times exp(\sum_{n=1}^{N}W_{n}\times \log P_{n})$$(15)$$Rouge-L=\frac{\left(1+\beta^{2}\right)R_{lcs}P_{lcs}}{R_{lcs}+\beta^{2}P_{lcs}}$$
where in the formula for BLEU, $P_{n}$ denotes the matching degree, $W_{n}$ is the weight of each n-gram with the same weight by default, and BP is a penalty factor. In the formula for Rouge-L, $R_{lcs}$, $P_{lcs}$ denote the recall and accuracy, respectively, and β is set to be a very large number so that Rouge-L considers $R_{lcs}$ almost exclusively.

### 4.2. Experiments

In terms of semantic segmentation, this article sets up comparison experiments to compare the image quality of the enhanced image with the original image and whether there is any improvement in the accuracy of semantic segmentation after data enhancement and adding a channel attention module. The comparison experiments are shown in Table 4.

After each complete training iteration, the model is evaluated for performance on the validation set to find the iterations that achieve optimal performance and prevent overfitting.

In terms of text generation, comparison experiments are set up to determine the effect of fine-tuning the model on the underground pipeline corridors dataset. The comparison experiments are shown in Table 5. After training is completed, two performance metrics are performed on the test set.

After the low light enhancement, the enhanced image is compared with the original image, and the corresponding three metrics Peak Signal-to-Noise Ratio(PSNR), Structural Similarity(SSIM), and Mean Absolute Error(MAE) are shown in Table 6. Larger values of PSNR and SSIM and smaller values of MAE mean a higher quality of the enhanced image. From the table, it can be seen that in terms of data enhancement of underground pipe corridors, the image enhanced by the Zero-DCE network has a higher image quality and is more suitable for the subsequent semantic segmentation task. The data of the original underground pipeline corridors and after performing the low light enhancement are shown in Figure 12.

As can be seen in Figure 12, the visibility of the underground pipeline corridors image data enhanced by the low-light enhancement algorithm becomes higher, and the feature information that can be extracted from the same photo after low-light enhancement increases, which also lays a good foundation for the subsequent semantic segmentation task.

As can be seen in Figure 13, in the enhanced image, the red (noise) in the flame area not only covers a larger area but also exhibits a denser texture. This directly indicates that after enhancement, the number of noisy pixels in the flame area has increased, and the noise fluctuations have become more intense.

The total loss of the semantic segmentation model during training is shown in Figure 14. After 160,000 iterations, the loss of the model is minimized and the performance is optimal.

Comparing the semantic segmentation accuracies of the low-light-enhanced data and the original data, the segmentation accuracies of the two batches of data are shown in Table 7.

From Table 7, it can be seen that after enhancement, except for a very few objects whose segmentation accuracy has decreased, the segmentation accuracy of most of the objects is improved, and the overall segmentation accuracy is also improved. Low-light enhancement of the image can effectively increase the feature information, and allow the model to extract more feature information, which in turn improves the semantic segmentation accuracy of the model. With the addition of the channel attention module, the model is able to focus its attention on certain channels that are more important for distinguishing the categories, which in turn improves the semantic segmentation accuracy. The original network achieves the highest segmentation accuracy on the enhanced image after adding the channel attention module. Category fluctuations are all below the 10% engineering threshold, representing local trade-offs resulting from module optimization rather than model instability.

The negative growth in a few categories is minimal, stems from reasonable model optimization trade-offs, and does not impact core functionality (overall performance improvement). This falls entirely within acceptable engineering parameters, requiring no additional adjustments to the model architecture. The current performance already meets the practical requirements for utility tunnel inspections.

As shown in the experimental results of Table 7, the Zero-DCE + DCNv4 + SE fusion model exhibits a certain degree of decline in IoU performance for segmentation tasks involving two target categories: “Flame” and “Gas Detector.” Specifically, its IoU for the flame category is 79.47%, lower than the 81.7% achieved by the original DCNv4 model. while the IoU for gas detectors was 92.59%, below the 93.27% achieved by the Zero-DCE + DCNv4 model. This performance fluctuation stems not from a single module failure, but from the combined effects of insufficient feature adaptability and inadequate category-specific matching during the fusion process. The specific causes and potential improvements are analyzed below.

One core contributing factor is the imbalanced feature weight distribution within the SE module. The SE module employs a “Squeeze-Excitation” mechanism to adaptively calibrate channel feature weights, whose performance relies on accurately identifying and amplifying key target features. However, in this experiment, it exhibited an adaptation bias toward the category features of flames and gas detectors. Flame targets exhibit blurred edges, irregular textures, and dispersed pixel distributions, while gas detectors typically appear as small objects with regular shapes and uniform textures. Both are easily obscured in images by more dominant background features. During global feature compression and activation, the SE module may disproportionately allocate weights to background or dominant categories, thereby suppressing channel responses for critical features like flame edges and gas detector contours. This reduces the model’s sensitivity to extracting these target features, ultimately lowering the IoU.

Addressing these issues, the model presents clear room for improvement across three dimensions: module optimization, collaborative strategies, and data-loss adaptation. First, optimize the SE module’s category adaptability by replacing the generic SE module with a category-aware variant. Train channel excitation weights separately for flames and gas detectors, while simplifying the fully connected layer structure and introducing Dropout to suppress noise, preventing imbalanced feature weight distribution. Second, establish a feature adaptation bridge between Zero-DCE and DCNv4 by inserting a 1 × 1 convolution + Batch Normalization layer. This maps enhanced features to DCNv4’s feature space while incorporating an illumination intensity branch to adaptively enable and adjust Zero-DCE intensity, preventing ineffective enhancement. Third, we enhance data and loss specificity by oversampling both target and non-target samples. Combining edge enhancement and random scaling strategies expands feature diversity. A Dice-IoU hybrid loss function increases the model’s focus on edges and small target regions, mitigating segmentation boundary inaccuracies.

When training with the LoRA method of fine-tuning, the number of parameters changed accounts for less than 1% of the number of parameters in the whole model, which makes it possible to train large models according to a specific task even with limited computational resources.

LoRA has a rank of 8. It will be injected into all core network layers of the Qwen2-VL-7B model that support LoRA adaptation, such as attention layers and feedforward network layers.

The training loss for this article, fine-tuning and training the visual–linguistic model, is shown in Figure 15. The training loss curve shows that the model fully converges after about 200 steps of iterations, at which point the performance metrics computed with this weight on the test set and the original model on the test set are shown in Table 8.

As can be seen from Table 8, the fine-tuned trained visual–linguistic model performs much better than the original model on the test set, with a huge improvement in both BLEU-4 (the weighted average of BLEUs computed from 1–4 g) and Rouge-L. This suggests that by fine-tuning the visual–linguistic model, we have improved the model’s comprehension accuracy for underground pipeline corridors scene understanding, allowing the model to generate more accurate and complete scene understanding text for our specific task.

Table 9 presents comparative experiments of different fine-tuning strategies across parameter count, training duration, and generation metrics. As shown, the LoRA method achieves high performance across all metrics, demonstrating the rationale for selecting this approach. Within the text templates, most content is predefined, while sections describing object interactions remain open-ended—reflecting a balance between fixed paradigms and diversity.

The original model achieved a BLEU-4 score of 1.08, primarily measuring the N-gram alignment between its generated utility tunnel descriptions and human-annotated reference texts. This extremely low score indicates the original model is entirely unsuitable for text generation tasks in underground utility tunnel scenarios. A score of 1.08 reflects the poor performance of a general-purpose model in this specific context, proving the original model cannot be directly applied to utility tunnel text generation. The score surged after fine-tuning, demonstrating that through LoRA fine-tuning, the model learned to recognize utility tunnel-specific objects and comprehend scene logic. The generated text achieved high alignment with the human-annotated reference, significantly improving adaptability and accuracy.

Our scene understanding system achieves single-frame inference in just 0.0673 s, fully meeting the real-time inspection requirements for utility tunnels. Regarding core specifications, equipment image capture typically operates at 30 frames per second. With an inference speed of 0.0673 s per frame, it precisely matches the capture rhythm, preventing image backlogs or critical frame omissions. For pipeline tunnel inspections, the end-to-end latency requirement for “detection → alert” of critical hazards like flames is less than or equal to 1 s. This inference speed occupies only 6.7% of the latency threshold, leaving ample time for subsequent alert triggering and manual response. It significantly outpaces the reaction speed of manual hazard inspections, fully meeting the response demands of emergency scenarios.

The visualization interface for scene understanding is shown in Figure 16. In the scene understanding visualization interface, the leftmost one is the original image of the current frame, the middle one is the semantic segmentation result of the image, and the rightmost one is the scene understanding text generation result of the image. The figure shows the scene understanding results of the three frames before and after a scene video after frame-splitting. During the scene understanding process of the first image frame, the visual–linguistic model generates wrong text with incorrect locations of the worker in the text, and the wrong text descriptions can be corrected efficiently by utilizing the category and location information of the objects in the image segmentation result map. The colored parts of the scene understanding text are the text parts that are differentiated in the three frames of the image: the red parts are the incorrect text generated by the visual–linguistic model, and the green parts are the correct text. The bottom section displays scene understanding visualizations for different compartments.

This study offers broad and practical applications, with its core value lying in providing an integrated intelligent solution for underground utility tunnel operation and maintenance. This solution combines “low-light adaptation + precise segmentation + standardized text generation.” This scenario-customized technical framework and low-cost model adaptation scheme can be seamlessly deployed to underground infrastructure such as cable tunnels and subway corridors, as well as high-risk confined spaces like chemical industrial parks and mine tunnels. It provides critical support for digital management of urban underground spaces, upgrading intelligent inspection equipment, and implementing multimodal perception technologies.

## 5. Discussion

The Zero-DCE+DCNv4 + SE fusion model adapts to low-light, high-noise scenarios in utility tunnels, achieving an mIOU of 82.39%—a 0.95% improvement over the baseline DCNv4—validating the synergistic effect of low-light enhancement and channel attention mechanisms. The Qwen2-VL-7B large model, after structured template annotation and LoRA fine-tuning, achieved a BLEU-4 score increase from 1.08% to 74.74% and a Rouge-L improvement from 9.29% to 84.85%. It can follow template descriptions to depict scenes and correct textual positioning errors using segmentation results. The system employs a pipeline parallel processing mechanism to adapt to image acquisition rates. For end-to-end latency requirements in critical hazard detection and alarm systems, the total delay—including single-frame inference time plus transmission and alarm processing—remains well below safety standards. Real-time performance stems from an engineered lightweight architecture: DCNv4 maintains controllable computational complexity, while Zero-DCE employs a 7-layer lightweight convolutional network. LoRA fine-tuning affects less than 1% of model parameters, precisely matching the drone’s 0.6m/s inspection speed and 6m detection field of view.

During training of scene understanding models, the validation loss curve is not recorded. If the training loss continues to decrease while the validation loss keeps increasing, it may indicate model overfitting. In such cases, overfitting should be mitigated by increasing the dataset size or reducing model complexity. Compared to the latest mainstream low-light enhancement models, images processed with Zero-DCE enhancement exhibit two primary drawbacks. Firstly, they carry a high risk of color distortion. In low-light images with significant noise, Zero-DCE amplifies brightness while simultaneously amplifying color noise, further exacerbating color shifts. Secondly, they demonstrate insufficient detail preservation. When handling low-light + high-noise images, they face the conflicting challenge of either over-smoothing details or amplifying noise.

When the illumination intensity in the duct falls below the critical threshold, the Zero-DCE enhancement effect becomes ineffective, leading to inaccurate semantic segmentation feature extraction and an increase in the false negative rate for small objects. For categories with extremely small sample sizes, segmentation accuracy may plummet in complex scenarios, compromising the reliability of hazard detection.

Future research will expand to cover more types of utility tunnels and special scenarios, refining the scenario classification system. Integrate data from gas sensors, temperature sensors, humidity sensors, and other sources to build a visual + sensor multimodal scene understanding framework, enhancing the accuracy and comprehensiveness of hazard identification.

## 6. Conclusions

This paper proposes a scene understanding system for underground pipeline corridor scenes. Firstly, this paper proposes an object recognition method based on semantic segmentation of low-light-enhanced images. Secondly, analyzing image data and fine-tuning a visual–linguistic model in order to generate scene description texts. Thirdly, by integrating the above functions into the system, the system is able to promptly process real-time captured images and generate scene understanding results. The results show that the accuracy of semantic segmentation on our model is improved. The fine-tuned visual–linguistic model has great improvement in all aspects of the metrics. In the future, we will continue to improve the scene understanding system by adding gas sensor data, so that the system can provide a more accurate and comprehensive scene understanding results.

## Figures and Tables

**Figure 1 sensors-26-00141-f001:**
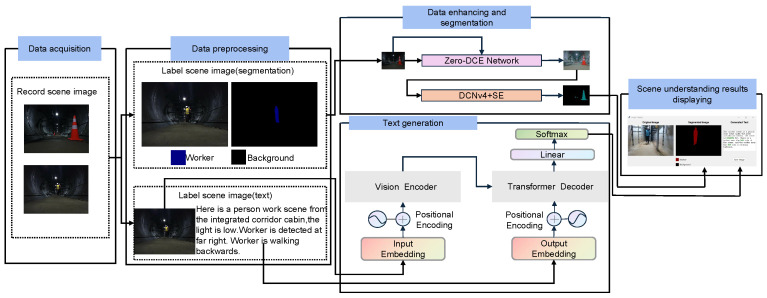
System architecture diagram.

**Figure 2 sensors-26-00141-f002:**
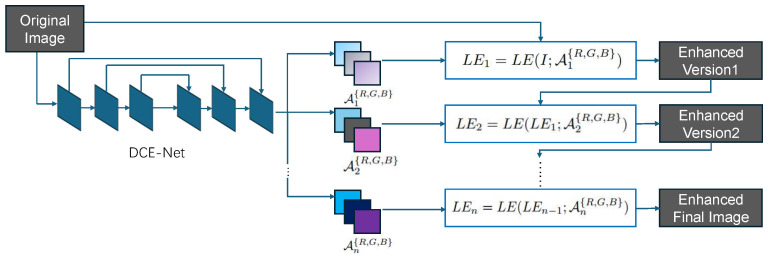
Overall framework of Zero-DCE network.

**Figure 3 sensors-26-00141-f003:**
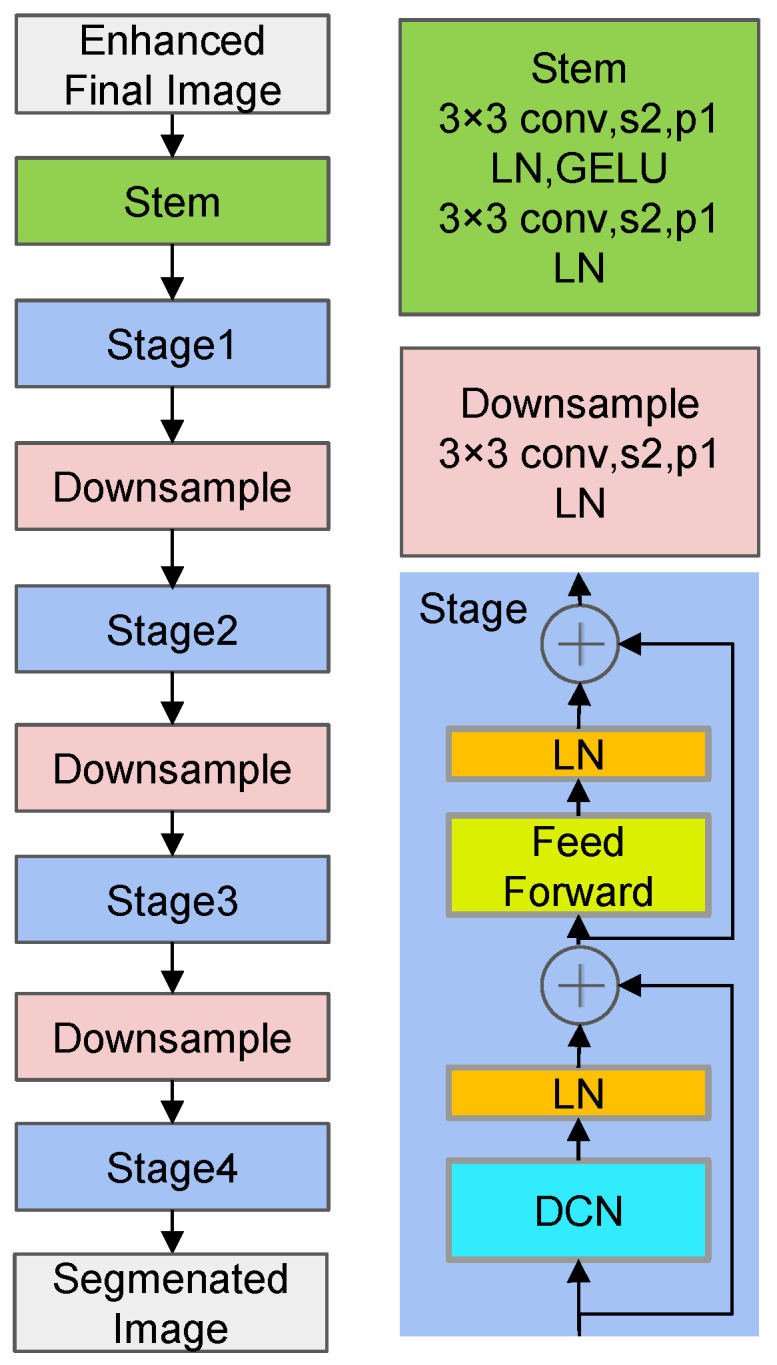
The overall structure of DCNv4 network.

**Figure 4 sensors-26-00141-f004:**
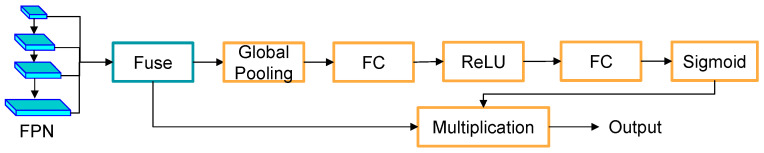
Structural diagram of adding SE module.

**Figure 5 sensors-26-00141-f005:**
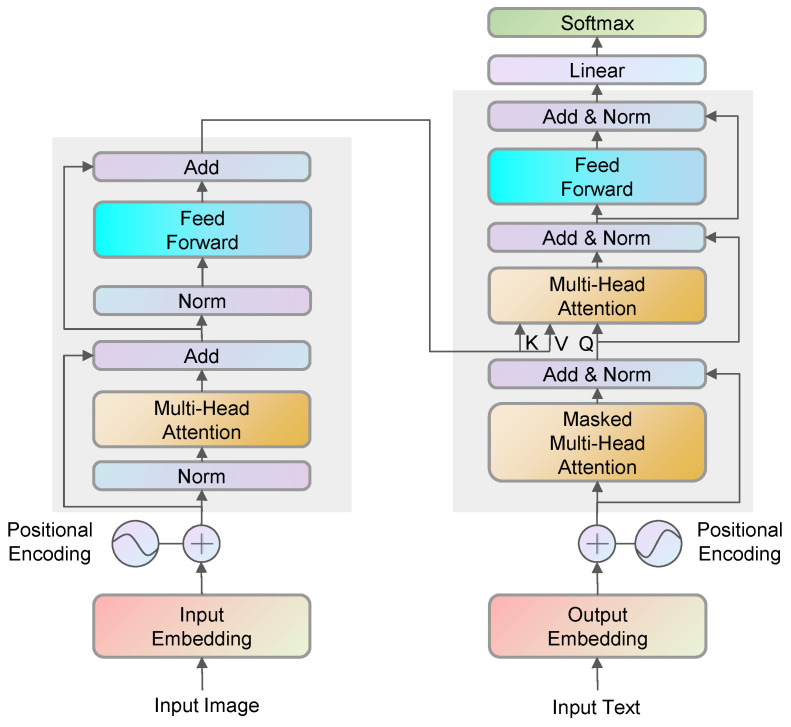
Architecture of visual–linguistic model.

**Figure 6 sensors-26-00141-f006:**
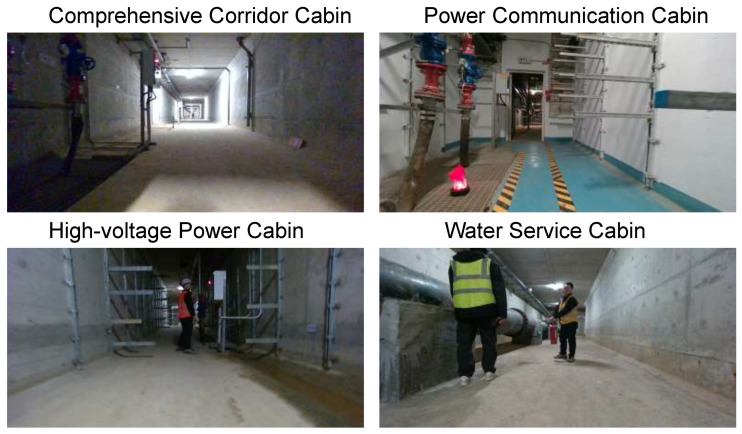
Different cabins in real underground pipeline corridors environment.

**Figure 7 sensors-26-00141-f007:**
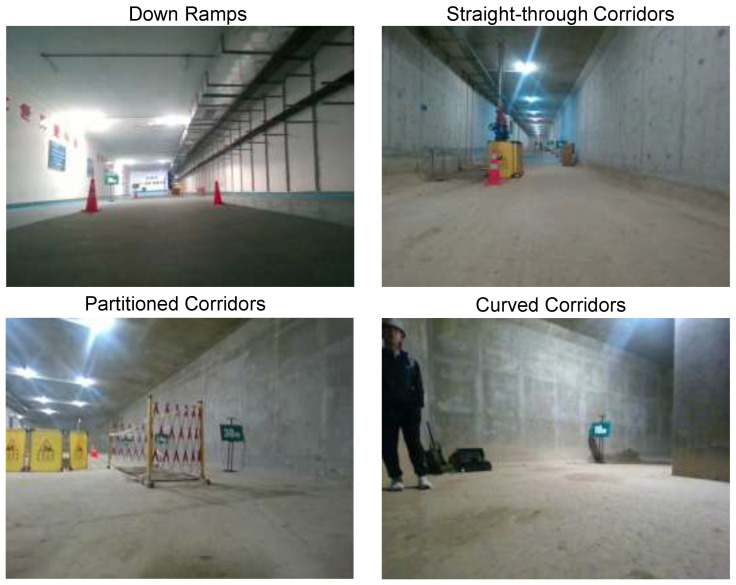
Different irregular spaces in real underground pipeline corridors environment.

**Figure 8 sensors-26-00141-f008:**
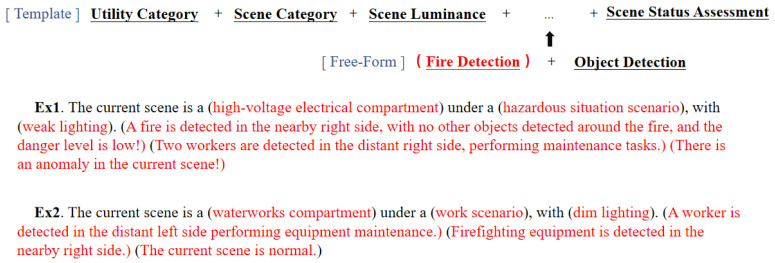
Text annotation templates.

**Figure 9 sensors-26-00141-f009:**
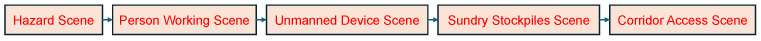
Scene category.

**Figure 10 sensors-26-00141-f010:**
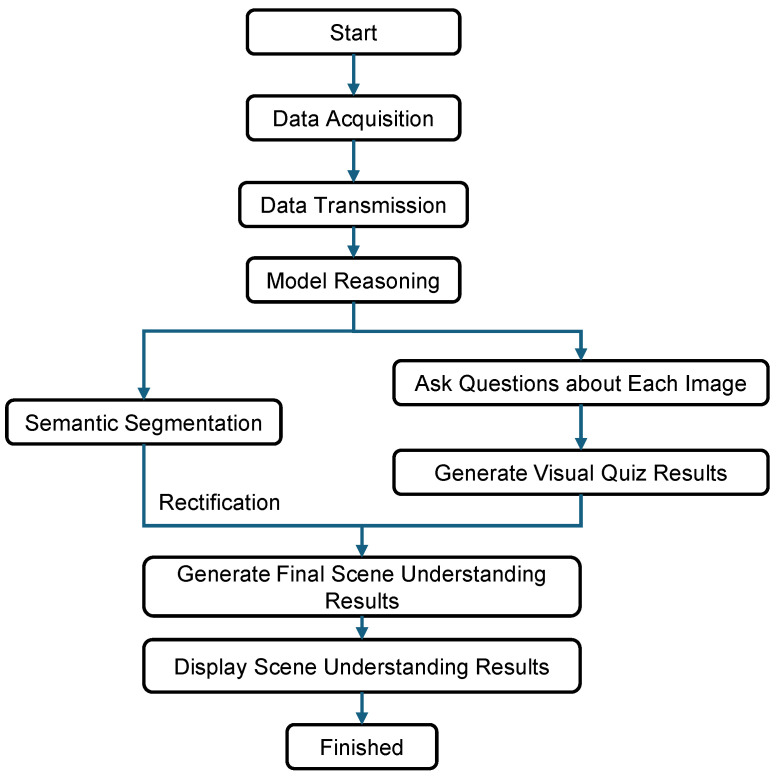
Scene understanding visualization flowchart.

**Figure 11 sensors-26-00141-f011:**
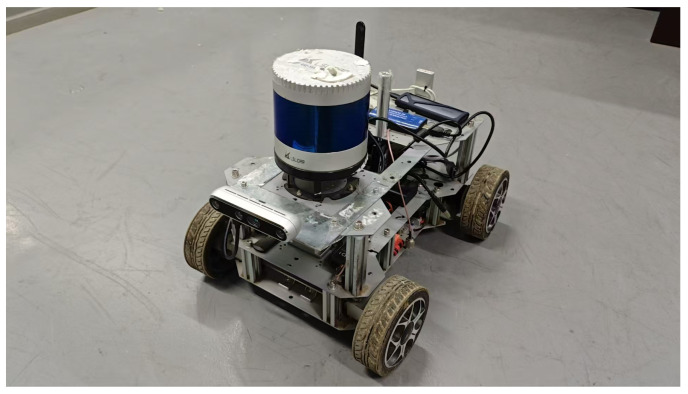
Unmanned vehicle.

**Figure 12 sensors-26-00141-f012:**
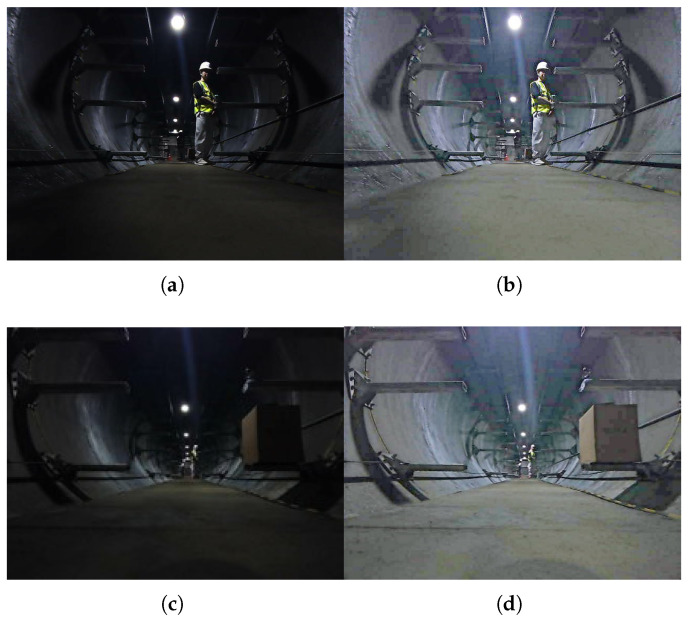
(**a**) Image before enhancement in the first scene; (**b**) Enhanced image of the first scene; (**c**) Image before enhancement in the second scene; (**d**) Enhanced image of the second scene.

**Figure 13 sensors-26-00141-f013:**
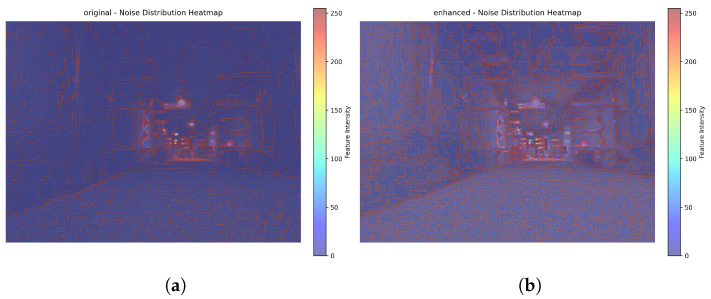
(**a**) Visualization before enhancement; (**b**) Visualization after enhancement.

**Figure 14 sensors-26-00141-f014:**
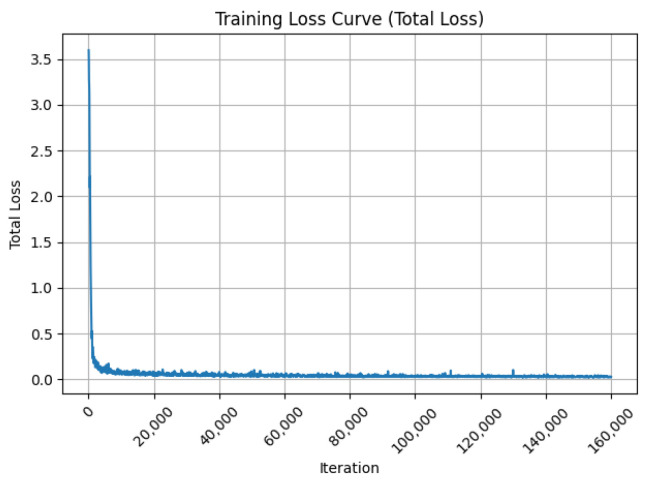
Total loss of the semantic segmentation model during training.

**Figure 15 sensors-26-00141-f015:**
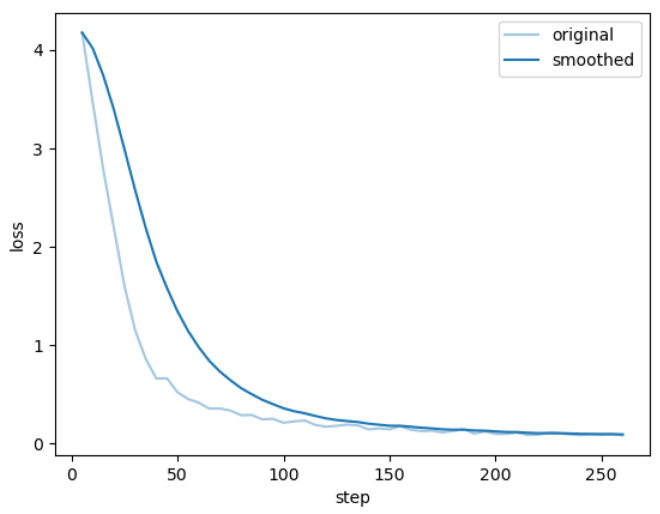
Loss profiles for fine-tuned training of visual-verbal model.

**Figure 16 sensors-26-00141-f016:**
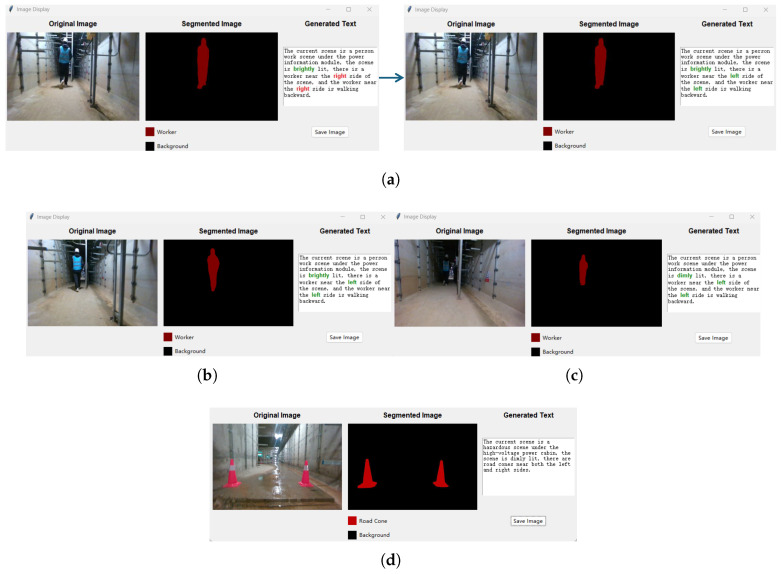
(**a**) Scene understanding results before and after first-frame error correction; (**b**) Scene understanding results for the second frame; (**c**) Scene understanding results for the third frame; (**d**) Scene understanding results for other compartments.

**Table 1 sensors-26-00141-t001:** Semantic segmentation dataset.

Class	Amount	Class	Amount
Background	10,917	Road Cone	9620
Worker	6054	Gas Detector	3500
Box	9483	Camera	3655
Rat	6579	Distribution Box	4369
Pressure Gauge	6707	Safety Indicator	6548
Flame	3216	Warning Post	5310
Road Pit	7906	Construction Fence	6064
Shovel	7727		

**Table 2 sensors-26-00141-t002:** Overview of the utility tunnel dataset.

Attribute	Details	Training	Testing
Total Images	6819	6149 (90%)	670 (10%)
Data Collection Period	May 2025–June 2025	A week in Wuhan’s tunnels	
Sampled Corridor Locations	15 irregular spaces across 4 cabin types	Comprehensive corridor 2069 (30.34%) Power communication 2801 (41.08%) High-voltage power 1033 (15.15%) Water service 916 (13.43%)	
Image Resolution	640 × 480	Unified Resolution	

**Table 3 sensors-26-00141-t003:** The configuration of unmanned vehicle.

	Unmanned Vehicle
Configuration	Ubuntu 20.04 D455 Camera
	Wireless Handle ROS System

Note: Manufacturer: Intel Corporation, Santa Clara, CA, USA.

**Table 4 sensors-26-00141-t004:** Semantic segmentation comparison experiments.

	DCNv4 DCNv4 + SE Zero-DCE + DCNv4 Zero-DCE + DCNv4 + SE
Environment	python 3.7 CUDA 11.3 cuDNN 8.2
	pytorch 1.11 NVIDIA RTX 3090 GPU
Hyperparameters	Adam optimizer initial learning rate 0.0006
	weight decay 0.05 batch size 2

**Table 5 sensors-26-00141-t005:** Text generation comparison experiments.

	Qwen2-Original Qwen2-Finetuned
Environment	python 3.9 CUDA 12.1 cuDNN 8.9
	pytorch 2.1 NVIDIA RTX 4090 GPU
Hyperparameters	Adam optimizer initial learning rate 0.0001
	maximum truncation length 2048 batchsize 2

**Table 6 sensors-26-00141-t006:** Comparison of image quality metrics after image enhancement by different methods.

Method	PSNR	SSIM	MAE
LIME [44]	9.61	0.22	80.42
EnlightenGAN	9.64	0.25	77.43
Zero-DCE	11.16	0.29	67.29

**Table 7 sensors-26-00141-t007:** Segmentation accuracy of models.

Class		DCNv4	DCNv4 + SE	Zero-DCE + DCNv4	Zero-DCE + DCNv4 + SE
Background	IOU	99.24	99.25	99.27	99.28
Worker	IOU	84.52	84.78	85.44	85.24
Box	IOU	87.28	87.15	87.85	88.0
Rat	IOU	79.62	80.06	80.4	80.24
Pressure Gauge	IOU	61.13	60.7	62.31	63.25
Flame	IOU	81.7	77.09	80.31	79.47
Road Pit	IOU	83.32	84.27	83.13	85.4
Shovel	IOU	64.99	66.38	66.89	67.44
Road Cone	IOU	86.65	87.05	87.1	87.13
Construction Fence	IOU	91.65	92.36	91.72	92.17
Warning Post	IOU	88.86	88.89	89.37	89.19
Safety Indicator	IOU	86.33	86.45	86.75	86.55
Distribution Box	IOU	84.62	84.95	84.75	85.62
Camera	IOU	49.08	52.62	52.26	54.32
Gas Detector	IOU	92.6	90.32	93.27	92.59
	mIOU	81.44	81.49	82.05	82.39

Note: All IoU values are the mean of 3 independent experiments, with the standard deviation of all classes ≤ 0.03, indicating high stability of the segmentation results.

**Table 8 sensors-26-00141-t008:** Performance metrics of models.

Model	BLEU-4	Rouge-L
Qwen2-VL-7B-Original	1.08	9.29
Qwen2-VL-7B-Finetuned	74.74	84.85

**Table 9 sensors-26-00141-t009:** Comparative experiments of different fine-tuning methods.

Fine-Tuning Method	Parameters	Training Time	Precision
Freeze	466,115,584	281 h	67.8
LoRA	20,185,088	35 h	74.74

## Data Availability

The data presented in this study are not publicly available.

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
