# Peer review of "Scene Understanding System of Underground Pipeline Corridors Under Characteristic Degradation Conditions"

_sensors, 2025, doi:10.3390/s26010141_

Round 1

Reviewer 1 Report

Comments and Suggestions for Authors

This work integrates semantic segmentation, visual-linguistic text generation, and low-light image enhancement to address scene understanding in subterranean pipeline corridors. Even though the application domain is pertinent and the system integration seems functional, the work needs significant improvements in experimental rigor, methodological transparency.

Abstract uses vague claims such as "has improved", "great improvement", without quantitative results. Replace vague improvement claims with specific numbers and ensure consistency between abstract and conclusion. Also, add more relevant keywords.

Concept diagram should be added in the introduction section.

The specific methodological innovations are not explicitly described in the manuscript. The authors are advised to clarify the innovative scientific or technological contributions that distinguish this study from existing methodologies. Either demonstrate algorithmic innovations or reframe as an applications paper with comprehensive validation.

Labels of figure 1 should be clear. Higher resolution would benefit from showing relevant details.

Dataset characteristics are described incompletely at Lines 278-279 and Table 1. Add a new table for the dataset information including all the details such as number of images, train/validation/test split (numbers and percentages), data collection timeframe, and number of different corridor locations sampled. Clarify in Section 4.1 whether data comes from one corridor or multiple corridors. If from single location, acknowledge this as a limitation and discuss implications for generalization.

Table 4 and 7 lack statistical validation for the performance comparisons. Please add standard deviations or confidence intervals to the Table 4 results. For Table 7, clarify what the BLEU-4 score of 1.08 for the original model represents. Include a brief statistical analysis section discussing the significance of the reported improvements.

Section 3 would benefit from additional implementation details. Specify the value of E in Equation 4, report the values of W_col and W_tvA in Equation 2. Describing the training procedure, including iterations and convergence criteria.

For Semantic Segmentation, mention how class imbalance is addressed, and explain the rationale for choosing DCNv4 over other possible architectures. For Text Generation clarify the LoRA rank, the layers adapted, and the number of training samples. Explain how segmentation outputs are used for rectification.

In Figures 13 and 14, include validation loss curves. Also state whether any signs of overfitting were observed and how overfitting was monitored or mitigated. If validation data was not recorded during training, this limitation should be clearly acknowledged in the manuscript.

Figure 15 shows limited system demonstration as it is only a single example. Would be possible to add more examples from different environments or lighting conditions?

Please explain whether templates are used only for training annotation or also during inference, and whether the model outputs template-structured or free-form text.

Incorporate recent enhancement methods in Table 6 for baseline comparison or acknowledge this limitation in the Discussion with a brief justification.

Add a discussion subsection covering limitations, failure modes, and future work would provide a more complete narrative.

Reviewer 2 Report

Comments and Suggestions for Authors

The comments have been attached.

Round 2

Reviewer 1 Report

Comments and Suggestions for Authors

Authors have addressed the previous comments carefully. However, there is a major structural concern that must be addressed. 

Add discussion section after the results section, and it should include performance analysis, limitations, failure modes and future work.

Conclusion should be the last section.

Reviewer 2 Report

Comments and Suggestions for Authors

 The originality or novelty should be described more clearly, and the significance of this research in any application areas can be illustrated in detail. The English can be improved, several Language polishing companies by MDPI are suggested. 
